# Neural Network-Augmented Locally Adaptive Linear Regression Model for Tabular Data

**Lkhagvadorj Munkhdalai [1], Tsendsuren Munkhdalai [2], Pham Van Huy [3], Jang-Eui Hong [4], Keun Ho Ryu [3,5,*] and Nipon Theera-Umpon [5,6]**

[1] Database and Bioinformatics Laboratory, College of Electrical and Computer Engineering, Chungbuk National University, Cheongju 28644, Republic of Korea
[2] Google, Mountain View, CA 94043, USA
[3] Data Science Laboratory, Faculty of Information Technology, Ton Duc Thang University, Ho Chi Minh City 700000, Vietnam
[4] Software Intelligence Engineering Laboratory, Department of Computer Science, Chungbuk National University, Cheongju 28644, Republic of Korea
[5] Biomedical Engineering Institute, Chiang Mai University, Chiang Mai 50200, Thailand
[6] Department of Electrical Engineering, Faculty of Engineering, Chiang Mai University, Chiang Mai 50200, Thailand
[*] Correspondence: khryu@tdtu.edu.vn or khryu@ieee.com; Tel.: +82-43-267-225; Fax: +82-43-275-2254

**Abstract:** Creating an interpretable model with high predictive performance is crucial in eXplainable AI (XAI) field. We introduce an interpretable neural network-based regression model for tabular data in this study. Our proposed model uses ordinary least squares (OLS) regression as a base-learner, and we re-update the parameters of our base-learner by using neural networks, which is a meta-learner in our proposed model. The meta-learner updates the regression coefficients using the confidence interval formula. We extensively compared our proposed model to other benchmark approaches on public datasets for regression task. The results showed that our proposed neural network-based interpretable model showed outperformed results compared to the benchmark models. We also applied our proposed model to the synthetic data to measure model interpretability, and we showed that our proposed model can explain the correlation between input and output variables by approximating the local linear function for each point. In addition, we trained our model on the economic data to discover the correlation between the central bank policy rate and inflation over time. As a result, it is drawn that the effect of central bank policy rates on inflation tends to strengthen during a recession and weaken during an expansion. We also performed the analysis on $CO_2$ emission data, and our model discovered some interesting explanations between input and target variables, such as a parabolic relationship between $CO_2$ emissions and gross national product (GNP). Finally, these experiments showed that our proposed neural network-based interpretable model could be applicable for many real-world applications where data type is tabular and explainable models are required.

**Keywords:** interpretable model; linear regression; neural network; adaptive learning; tabular data; economic management; environmental economics

## 1. Introduction

Tabular data are the most common type because it covers many exciting problems in various domains. In addition, the predictive and explanatory modeling on tabular data is a non-trivial task because those models often need to be interpreted by explaining real-world phenomena.

Artificial intelligence (AI) recently has shown super-human performance in many domains including image processing, natural language processing, etc. However, researchers have been developing very complex black box models to achieve the high predictive performance. Nonetheless, such complex deep learning models do not show good

predictive accuracy on tabular data, and it is challenging to explain output of them. Therefore, simple interpretable machine learning models are still broadly applied for modeling tabular data. For example, ordinary least squares (OLS) regression has been extensively employed to explain a wide variety of economic relationships [1,2]. Because the statistical properties of linear regression make it trustworthy, linear regression coefficients have been used as a model interpreter by determining the effect of each input feature on the output [3]. Unfortunately, the predictive capacity of OLS regression is not stronger than black box machine learning models [4]. On the other hand, deep learning models achieved significantly higher predictive performance on those types of data, such as audios, images or videos, and texts. Still, they have not shown better predictive performance than ensemble models, such as lightGBM, CatBoost, etc., on tabular data [5,6].

In addition, many techniques for explainability in machine learning (ML) have been proposed to understand the predictions provided by complex ML models. The most popular methods for explainable AI are shapley additive explanations (SHAP) and local interpretable model-agnostic explanations (LIME) [7,8]. Unfortunately, researchers have not done much work designing explainable methods for tabular data [9].

In this study, we introduce a novel neural network-based locally adaptive linear regression model by combining a simple OLS regression and feed-forward neural networks to provide model interpretability with high predictive performance for tabular data. Our proposed framework for tabular data is shown in Figure 1. We first estimate the linear regression coefficients using OLS method to generalize our base learner. In this phase, we do not normalize the input data ($x$) to obtain meaningful regression coefficients to calculate the exact effect of each input variable on the output. In the second phase, we train neural network model based on the normalized input data ($x'$) to update the regression coefficients for each current observation. The neural networks are used as our meta-learner to adjust the weight parameter of base-learner as known as regression coefficients for each observation to improve the predictive performance. As shown in Figure 1, we update each regression coefficient using the formula for the confidence interval (CI) and then rebuild an interpretable local linear model for each observation.

In our proposed model, since we use the formula for a CI, the adjusted regression coefficients should be in a range between their lower and upper confidence intervals. Therefore, our proposed model can avoid overfitting, and the model interpretable is identical with OLS regression. At the same time, we can extend the predictive power of linear regression.

We evaluated the predictive performance and model interpretability of our proposed approach on the benchmark tabular datasets for regression task. Our proposed model achieved slightly higher predictive performance than regression and the state-of-the-art models. Furthermore, we also showed that our proposed model can measure a local effect of each input variable on the output for each observation.

The contributions of this study are listed as follows:

(1) We introduce a neural network-based interpretable local linear regression model for tabular data.

(2) The linear regression coefficients are the same for all observations in a data, further degrading its predictive performance. We design regression coefficients as locally adaptable within their confidence intervals using a meta-level neural network model. Our proposed model parameterizes a local linear function for each example in a given data; therefore, the predictive performance of OLS regression is significantly improved.

(3) Our proposed model can avoid overfitting because the adapted coefficients range between their lower and upper confidence intervals.

(4) Our proposed model can measure a local effect of each input variable on the output. Therefore, our model can be applicable for many real-world applications, such as economics, biology, management, and social science, where data type is tabular and interpretable models are required.

The work is organized as follows. Section 2 presented the discussion of related research, and Section 3 introduced the proposal of our neural network-based architecture. The comparison and experiment results are presented in Section 4. We summarized this work and described the further research area at the end.

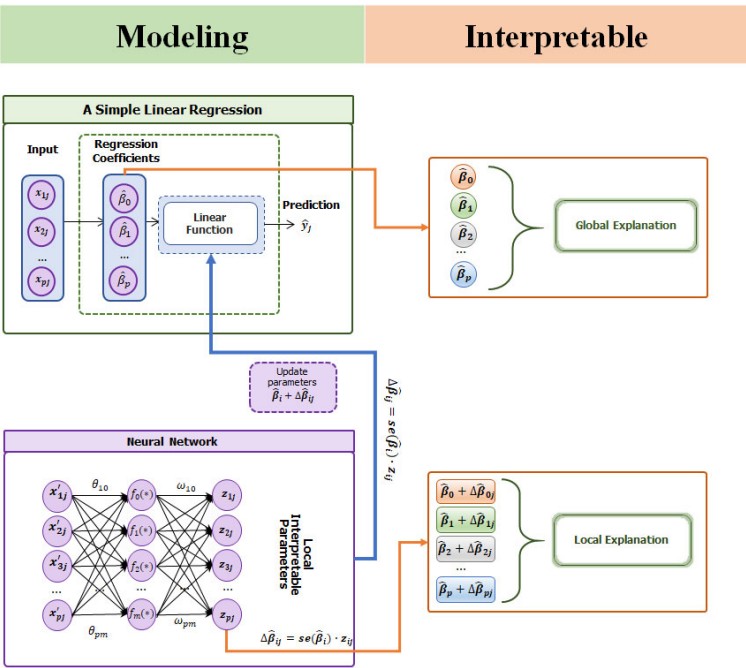

**Figure 1.** Summary of our proposed model, where $x_{ij}$ is *i-th* variable of *j-th* observation, $x'_{ij}$ is *i-th* normalized input variable of *j-th* observation, $\hat{y}_j$ is the *j-th* predicted output, $\hat{\beta}_i$ is estimated regression coefficient of *i-th* variable, se($\hat{\beta}_i$) is standard error of the *i-th* estimated regression coefficient, and $z_{ij}$ is predicted critical value for *i-th* variable of *j-th* observation. We first perform OLS estimator, and the resulting linear regression coefficients and standard errors are given as input to the neural network model. Second, we use feed-forward neural networks to adjust each regression coefficient using normalized input data. Finally, we reconstruct our local linear regression model for each observation based on the adjusted regression coefficients.

## 2. Related Work

Developing locally adaptive regression approaches have begun much earlier [10–13]. Those approaches can be classified into three main categories—nearest neighbor regression, weighted averaging regression, and locally weighted regression [14]. Nearest neighbor regression uses *k* most relative instances for a query instance to obtain its best fitting function for the target point [12,15]. Weighted averaging methods compute a weighted output of neighboring samples, weighted by using their similarities to the target point [16,17]. Locally weighted regression (LWR) is very similar to our proposed model because these models do not consider a fixed set of parameters for each instance [11,12]. However, the main weakness of nearest neighbor-based models is similar to the memory-based learning algorithms, where they keep full training data to provide the prediction for test data. This disadvantage makes them computationally expensive on large data. Instead, we use the neural networks as our meta-learner model that can adjust the parameters of the base-learner for each instance during the training process; therefore, our proposed model will be more efficient in terms of memory usage and time consumption on large datasets.

Using neural networks (meta-learner) to generate weight parameters for another one (base-learner) has been designed in the meta-learning field [18–20]. Based on this idea, we

train a meta-learner model to retrieve the local best linear model for each instance by producing the parameters of base-learner. The parameters produced by the meta-learner are known as fast-weights in the field of meta-learning. Our proposed model is similar to the Meta Networks (MetaNet) [21] that uses an additional neural network model to generate the fast-weights for rapid generalization. This approach has been successfully used on images, text, and audio data [22–24]. However, our proposed architecture is interpretable and for tabular data, which is different from MetaNet.

Furthermore, there are several similar approaches for tabular data [25–27]. For example, Bildirici and Ersin (2009) [25] improved GARCH family models by artificial neural networks for financial time-series data. Furthermore, Bildirici and Ersin (2014) [26] also combined Markov switching ARMA-GARCH model with neural networks to predict exchange rates and stock returns. They used neural network approach to predict the parameter of ARMA and GARCH models, which is very similar with our work. Recently, LocalGLMnet architecture has been proposed in Richman and Wuthrich, (2021) [27], which is also very similar with our proposed architecture. This model improves the predictive power of GLMs by using neural networks and provides an explainability same as GLMs. The authors predict the parameters of GLMs using neural network to achieve superior predictive performance. The idea of this work is same as our proposed model, but the predicted parameters cannot be consistency of explaining the logical relationship between input and output variables. This is because the predicted parameters of GLMs highly depend on the weight parameter initialization for neural networks since they use gradient descent optimization algorithm to train LocalGLMnet. In other words, the main advantage of our work is that we first perform OLS to obtain unbiased regression coefficients, and then update them into their confidence intervals using neural network approach.

Another similar work is done by Takagi and Sugeno [28] called the TS fuzzy model. This method offers a new technique to build multi-models representing local input–output relations of a non-linear system. However, due to a large number of variables and the nature of the continuous variables for the regression task, the TS fuzzy model usually utilizes a tremendously enormous number of rules and does not consider the complexity of the model [29]. Unlike TS fuzzy models, the benefit of our proposed model is that we do not use rules, and our meta-learner learns these rules automatically based on a given data.

In addition, although the state-of-the-art ML models have showed magnificent predictive performance in various domains, the inability to explain them decreases humans' trust. Subsequently, eXplainable AI (XAI) has become a significant and active research area [30]. Recently, a large number of studies has been done to understand the black-box model and increase humans' trust. However, most studies focused on post-explainability rather than explainable models [5,8,31,32].

## 3. Methodology

In our proposed model, there are two main components: base-learner and meta-learner (see Figure 1). In the first component, we obtain our interpretable base-learner using OLS regression on training data. We then train our meta-learner based on the neural network model using normalized training data to predict the changes of each regression coefficient for each observation. Finally, we construct our local interpretable model using adjusted local regression coefficients to predict the target point.

### 3.1. Ordinary Least Squares

We choose the most interpretable OLS regression as our base-learner model and present its theoretical and statistical properties in this section.

Given a set of data $(x_1, y_1), \dots (x_n, y_n)$ of $n$ observations and $p$ independent variables $(x_{i1}, \dots x_{ip})$, the OLS regression model is formulated as follows:

$$y_j = \beta_0 + \sum_{i=1}^{p} \beta_i x_{ij} + \varepsilon_j \tag{1}$$

where $y_j$ is the target sample of *j-th* observation, $x_{ij}$ is *i-th* variable for *j-th* observation, and $\varepsilon_j$ are random errors for *j-th* observation. OLS estimates the $\beta_i$ coefficients by minimizing the sum of squared residuals.

From Equation (1), the $\beta_i$ regression parameters can be obtained by the following formula:

$$\hat{\beta}_{OLS} = \left( X_n^{\top} X_n \right)^{-1} X_n^{\top} y_n \tag{2}$$

where $X_n = [x_1^{\top}, \dots x_n^{\top}] \in \mathbb{R}^{n \times p}$ is input data and $y_n = [y_1, \dots y_n] \in \mathbb{R}^n$ is target. However, we cannot estimate the true value of the regression coefficients from the sample data [1]. Therefore, the confidence interval for a regression coefficient in OLS regression is computed as shown below:

$$CI_{\beta_i} = \hat{\beta}_i \pm z_i \cdot se(\hat{\beta}_i) \tag{3}$$

where $z_i$ is the critical value and $se(\hat{\beta}_i)$ is the standard error of the coefficient $\hat{\beta}_i$.

### 3.2. Neural Networks

We use multi-layer perceptron (MLP) as a meta-learner in our proposed architecture [33,34]. The goal of the meta-learner model is to augment the base-learner to achieve better predictive performance. A simple MLP neural network architecture is contracted by input, hidden, and output layers. The hidden layer consists of a certain number of neurons and activation function.

The Equation (4) defines MLP neural network model consisting of two hidden layers with $N$ and $M$ nodes:

$$h_j^{(1)} = f_1(\sum_{i=1}^{p} \omega_{ji}^{(1)} x_i + w_{j0}^{(1)})$$

$$h_k^{(2)} = f_2(\sum_{j=1}^{N} \omega_{kj}^{(2)} h_j^{(1)} + w_{k0}^{(2)}) \tag{4}$$

$$\hat{y}_g = f_3(\sum_{k=1}^{M} \omega_{gk}^{(3)} h_k^{(2)} + w_{g0}^{(3)})$$

where $x_i$ is *i-th* the input variable of $p$ variables, $h_j^{(1)}$ and $h_k^{(2)}$ are the first and second hidden layers, $\omega_{ji}^{(1)}$ and $\omega_{j0}^{(1)}$ are the weight parameters for the first hidden layer, $\omega_{kj}^{(2)}$ and $\omega_{k0}^{(2)}$ are the weight parameters for the second hidden layer, $\omega_{gk}^{(3)}$ and $\omega_{g0}^{(3)}$ are the weight parameters for output layer, $\hat{y}_g$ is the *g-th* output and $f_1, f_2$ and $f_3$ are the activation functions for hidden and output layers.

### 3.3. The Proposed Model

Our model receives two types of inputs, non-normalized ($x$) and normalized ($x'$). The meta-learner takes normalized input to adjust the regression coefficients of the base-learner because the normalized input data help to learn the neural network models faster and achieve better predictive performance [35]. In contrast, the base-learner receives non-normalized input data to obtain an accurate explanation that is logically relevant to the real world because data scaling leads to misinterpreting the model [36].

As mentioned above, the input of the meta-learner is normalized input variables ($x'$) and the output should be the predicted $z$ critical value for each regression coefficient.

Since the $z$ critical value can be any number, we choose a linear function as an activation function of the output layer for meta-learner. Then our meta-learner for *i-th* variable is defined as follow:

$$z_i = \sum_{j=1}^{M} \omega_{ij}^m \cdot F_i(x', \theta_i^m) + \omega_{i0}^m \tag{5}$$

where $x'$ is normalized input, $z_i$ is the critical value for *i-th* variable, $\theta_i^m$ and $\omega_{ij}^m$ is the parameters of the meta-learner for *i-th* variable, $M$ is the number of nodes in the last hidden layer, and $F_i$ represents a neural network model without linear output layer for *i-th* variable.

Furthermore, a local linear regression model can easily be rebuild using the CI formula during the training phase. Then our proposed model for *j-th* observation is shown as follows:

$$y_j = \left( \hat{\beta}_0 + z_0(x'_j; \theta_0^m, \omega_0^m) \cdot se(\hat{\beta}_0) \right) + \sum_{i=1}^{p} \left( \hat{\beta}_i + z_i(x'_j; \theta_i^m, \omega_i^m) \cdot se(\hat{\beta}_i) \right) x_{ij} + \tag{6}$$

where $x_{ij}$ is *i-th* input variable of *j-th* observation, $x'_j$ is the normalized input variables of *j-th* observation, $z_i$ is the critical value for *i-th* variable, $se(\hat{\beta}_i)$ is the standard error of the *i-th* regression coefficient $\hat{\beta}_i$, and $\epsilon_j$ is error term of *j-th* observation.

### 3.4. Model Training

In Equation (7), $\hat{\beta}_i$ and $se(\hat{\beta}_i)$ are not learnable parameters, and $z_i$ contains learnable parameters, which are described in Equation (5). Therefore, we can perform the following transformation for *j-th* observation in Equation (6).

$$\begin{aligned} y_j &= \left( \hat{\beta}_0 + z_0(x'_j; \theta_0^m, \omega_0^m) \cdot se(\hat{\beta}_0) \right) + \sum_{i=1}^{p} \left( \hat{\beta}_i + z_i(x'_j; \theta_i^m, \omega_i^m) \cdot se(\hat{\beta}_i) \right) x_{ij} + \\ &= \left( \hat{\beta}_0 + \sum_{i=1}^{p} \hat{\beta}_i x_{ij} \right) + \left( z_0(x'_j; \theta_0^m, \omega_0^m) \cdot se(\hat{\beta}_0) + \sum_{i=1}^{p} z_i(x'_j; \theta_i^m, \omega_i^m) \cdot se(\hat{\beta}_i) x_{ij} \right) \end{aligned} \tag{7}$$

From Equation (7), we can distinguish the part that contains learnable parameters only. Then it can be written as follows:

$$y_j - \left( \hat{\beta}_0 + \sum_{i=1}^{p} \hat{\beta}_i x_{ij} \right) = z_0(x'_j; \theta_0^m, \omega_0^m) \cdot se(\hat{\beta}_0) + \sum_{i=1}^{p} z_i(x'_j; \theta_i^m, \omega_i^m) \cdot se(\hat{\beta}_i) x_i \tag{8}$$

$$e_j^{OLS} = z_0(x'_j; \theta_0^m, \omega_0^m) \cdot se(\hat{\beta}_0) + \sum_{i=1}^{p} z_i(x'_j; \theta_i^m, \omega_i^m) \cdot se(\hat{\beta}_i) x_{ij} + \varepsilon_j \tag{9}$$

where $e_j^{OLS}$ defines an unpredicted value by OLS for *j-th* observation. We can now formulate our loss function based on the Equation (9):

$$\begin{aligned} &\mathcal{L}\left( \theta_0^m, \theta_1^m, \dots \theta_p^m, \omega_0^m, \omega_1^m, \dots \omega_p^m \right) = \\ &\frac{1}{n} \sum_{j=1}^{n} \left| e_j^{OLS} - \left( z_0(x'_j; \theta_0^m, \omega_0^m) \cdot se(\hat{\beta}_0) + \sum_{i=1}^{p} z_i(x'_j; \theta_i^m, \omega_i^m) \cdot se(\hat{\beta}_i) x_{ij} \right) \right| \end{aligned} \tag{10}$$

We used the mean absolute error loss $(\mathcal{L})$ to optimize the parameters of meta-learner model $(\theta_i^m \ and \ \omega_i^m; i = 0, 1, \dots p)$.

Recall that our estimated regression coefficients and their standard errors are numerical input after performing OLS estimator. In addition, our meta-learner model can consist of one or multiple neural networks, and the output of meta-learner should be equal to the

number of variables. Both architectures can easily be trained with stochastic gradient descent (SGD) optimization with the backpropagation algorithm. The model training algorithm for our proposed model is as shown in algorithm 1. First, we perform OLS estimator to obtain regression coefficients, their standard error, and prediction of training and validation set as shown in line 1. We then start to train meta-learner model based on SGD optimization with the backpropagation algorithm from the line 3. In order to select the best model, early stopping algorithm is used. From line 11, we calculate the validation loss during training process for every epoch. Based on the patience number, which is criteria for early stopping algorithm, we select the best model as shown in the line 12–15.

---

**Algorithm 1.** Model training for our proposed model

---

**Input:** Training set non-normalized $\{x_i, y_i\}_{i=1}^N$ and normalized input $\{x_i'\}_{i=1}^N$
Validation set non-normalized $\{x_i, y_i\}_{i=1}^M$ and normalized input $\{x_i'\}_{i=1}^M$
Epoch number $num\_epoch$;
Early stopping patience number $max\_patience$;
Learning rate $\alpha$
**Output:** meta-learner $z(\theta^m, \omega^m)$, the estimated regression coefficients $\hat{\beta}$ and their standard errors $se(\hat{\beta})$

**Procedure: {**
1:   perform OLS estimator to obtain $\{\hat{\beta}; se(\hat{\beta})\}$; $\hat{y}^{train}$; $\hat{y}^{val}$
2:   $Loss_{val} = inf$; $e^{train} = y\text{-}\hat{y}^{train}$; $e^{val} = y\text{-}\hat{y}^{val}$; $patience = 0$
3:   **for** $ep=0$ **to** $epoch$ **do**                    //loop for *epoch* iterations
4:         **for** $j=1$ **to** $N$ **do**
5:             $\hat{e}_j = \left(z_0(x_j'; \theta_0^m, \omega_0^m) \cdot se(\hat{\beta}_0)\right) + \sum_{i=1}^p \left(z_i(x_j'; \theta_i^m, \omega_i^m) \cdot se(\hat{\beta}_i)\right)x_{ij}$
6:             $L_j \leftarrow loss_{train}(\hat{e}_j, e_j^{train})$
7:             $\theta_j^m \leftarrow \theta_{j-1}^m - \alpha\nabla_\theta L_j$          //gradient descent
8:             $\omega_j^m \leftarrow \omega_{j-1}^m - \alpha\nabla_b L_j$         //gradient descent
9:         **end for**
10:       $\hat{e}_{ep}^{val} = \left(z_0(x'; \theta_0^m, \omega_0^m) \cdot se(\hat{\beta}_0)\right) + \sum_{i=1}^p \left(z_i(x'; \theta_i^m, \omega_i^m) \cdot se(\hat{\beta}_l)\right)x_i$
11:       $L_{ep}^{val} \leftarrow loss_{val}(\hat{e}_{ep}^{val}, e^{val})$
12:       **if** $L_{ep}^{val} < Loss_{val}$ **do**
13:             $Loss_{val} = L_{ep}^{val}$
14:             $patience = 0$
15:             **return** $z(\theta^m, \omega^m), \{\hat{\beta}; se(\hat{\beta})\}$
16:       **else do**
17:             $patience = patience + 1$
18:       **if** $patience > pn$ **do**
19:             **break**
20:  **end for**
**}**

---

## 4. Experimental Result

### 4.1. Benchmark Datasets

We presented benchmark public tabular datasets in Table 1. These datasets for the regression task were employed to evaluate the predictive performance. Datasets from 1 to 5 were retrieved from UCI Machine Learning Repository for regression task [37]. Other 3 datasets are downloaded from different sources such as California Housing dataset for predicting house price is retrieved from [38], FICO dataset for credit scoring is download from [39] and Bodyfat for estimating body fat is downloaded from [40].

**Table 1.** Summary of datasets used in the comparison of the predictive performance.

| № | Dataset | Number of Observations | Number of Variables |
|---|---------|------------------------|---------------------|
| 1 | Concrete Strength | 1030 | 8 |
| 2 | Energy Efficiency | 768 | 8 |
| 3 | Naval Propulsion | 11,934 | 16 |
| 4 | Power Plant | 9568 | 4 |
| 5 | Protein Structure | 45,730 | 9 |
| 6 | California Housing | 20,640 | 9 |
| 7 | FICO | 10,459 | 20 |
| 8 | Bodyfat | 253 | 15 |

We also trained our model on synthetic and the real-world economic datasets to demonstrate the model interpretability in Section 4.4.

### 4.2. Baseline Models and Hyperparameters

For regression baselines, a linear regression (OLS), Bayesian (Bayesian) [41] lasso [42], and ridge [43] regressions are chosen in the predictive performance comparison.

For other alternative baselines, we compare our results to Neural Additive Model (NAM) [44] and TabNet [6] models, which are the most popular and high-performance interpretable models for tabular data. We also performed additional experiments using LightGBM [45] and Catboost [46] machine learning algorithms to compare their predictive performance to our model.

We also need to configure the model architecture of our meta-learner and its hyperparameters for training. We trained two types of architectures—meta-learner consists of multiple MLPs by assigning each MLP to each regression coefficient (mult MLP), and meta-learner consists of a single MLP with multiple outputs; number of neurons for output layer must be the same as the number of variables including intercept (single MLP). The architecture of each MLP is constructed by three hidden layers with {256, 256, and 256} neurons.

For other hyperparameters, we set the maximum epoch number equal to 10,000 and the learning rate equal to 0.01. We used an Early Stopping algorithm to select the best model on the validation set. We set the same configuration for all datasets and used the five-fold cross-validation method to evaluate and compare the models.

### 4.3. Evaluation Metrics

The evaluation metrics for regression task mostly calculate the error between the observed and predicted values for target variables [47]. The root mean square error (RMSE) and mean absolute error (MAE) are used to measure the model performances.

$$\text{RMSE} = \sqrt{\sum_{i=1}^{n} \frac{\left(\hat{y}_i - y_i\right)^2}{n}} \tag{11}$$

$$\text{MAE} = \sum_{i=1}^{n} \frac{\left|\hat{y}_i - y_i\right|}{n} \tag{12}$$

where $\hat{y}_i$ is the *i-th* predicted value, $y_i$ is the *i-th* observed value, and $n$ is the number of observation in the test set.

### 4.4. Comparison of Predictive Performance

The aim of this experiment analysis is to show how the predictive performance of OLS regression is improved after being augmented by neural networks.

Our proposed model achieved outstanding predictive performance on 3 out of 8 datasets (Energy Efficiency, Naval Propulsion, and Bodyfat) for RMSE evaluation metric and showed similar predictive performance on the other datasets (see Table 2). The regression baseline models showed poorer predictive performance than our proposed model on all datasets. As shown in Table 2, the predictive performance of OLS regression is weaker than that of the Lasso, Ridge, and Bayesian regressions, but its predictive power has improved significantly after being augmented by the neural networks. Table 3 reported the predictive performance on 8 datasets using the MAE evaluation metric. From the results, we can see that our proposed model notably outperformed the regression and the state-of-the-art baseline models on four datasets (Energy Efficiency, Naval Propulsion, Protein Structure, and Bodyfat).

**Table 2.** Results on benchmark datasets comparing RMSE.

| Model | Concrete Strength | Energy Efficiency | Naval Propulsion | Power Plant | Protein Structure | California Housing | FICO | Bodyfat |
|---|---|---|---|---|---|---|---|---|
| OLS | 10.527 ± 0.426 | 7.809 ± 0.549 | 0.016 ± 0.001 | 5.051 ± 0.161 | 5.191 ± 0.045 | 0.780 ± 0.025 | 6.081 ± 0.190 | 4.251 ± 1.124 |
| Lasso | 10.492 ± 0.452 | 7.199 ± 0.345 | 0.009 ± 0.001 | 4.559 ± 0.144 | 5.186 ± 0.044 | 0.726 ± 0.013 | 5.016 ± 0.080 | 1.224 ± 0.767 |
| Ridge | 10.534 ± 0.413 | 7.194 ± 0.346 | 0.007 ± 0.001 | 4.559 ± 0.145 | 5.185 ± 0.044 | 0.727 ± 0.015 | 5.02 ± 0.0820 | 1.274 ± 0.735 |
| Bayesian | 10.518 ± 0.439 | 7.590 ± 0.509 | 0.007 ± 0.001 | 4.560 ± 0.145 | 5.185 ± 0.044 | 0.727 ± 0.015 | 5.02 ± 0.0820 | 4.575 ± 0.598 |
| LGBM | **4.166 ± 0.698** | 0.547 ± 0.051 | 0.008 ± 0.001 | **3.145 ± 0.191** | 3.623 ± 0.055 | **0.438 ± 0.009** | 0.358 ± 0.074 | 1.858 ± 0.311 |
| CatBoost | 4.381 ± 0.708 | 0.371 ± 0.075 | 0.002 ± 0.003 | 3.330 ± 0.153 | **3.578 ± 0.057** | 0.445 ± 0.010 | **3.304 ± 0.066** | 1.662 ± 0.315 |
| Tabnet | 5.732 ± 0.712 | 3.608 ± 0.093 | 0.019 ± 0.003 | 0.427 ± 0.138 | 3.863 ± 0.035 | 0.629 ± 0.010 | 3.573 ± 0.079 | 4.478 ± 0.301 |
| NAM | 5.793 ± 0.875 | 3.180 ± 0.424 | 0.052 ± 0.027 | 3.689 ± 0.143 | 4.741 ± 0.041 | 0.5624 ± 0.007 | 3.490 ± 0.081 | 4.559 ± 0.68 |
| Ours (single MLP) | 5.007 ± 0.615 | 0.311 ± 0.073 | 0.001 ± 0.001 | 3.47 ± 0.127 | 3.905 ± 0.056 | 0.611 ± 0.097 | 0.391 ± 0.088 | **0.787 ± 0.37** |
| Ours (mult MLP) | 4.895 ± 0.616 | **0.303 ± 0.075** | **0.001 ± 0.001** | 3.633 ± 0.128 | 4.008 ± 0.067 | 0.615 ± 0.071 | 3.384 ± 0.093 | 0.808 ± 0.376 |

NA represents missing value that indicates the experiments cannot be performed because of the type of model and data size.

**Table 3.** Results on benchmark datasets comparing MAE.

| Model | Concrete Strength | Energy Efficiency | Naval Propulsion | Power Plant | Protein Structure | California Housing | FICO | Bodyfat |
|---|---|---|---|---|---|---|---|---|
| OLS | 8.346 ± 0.325 | 6.07 ± 0.374 | 0.013 ± 0.001 | 4.023 ± 0.078 | 4.325 ± 0.042 | 0.573 ± 0.010 | 4.440 ± 0.044 | 3.246 ± 0.398 |
| Lasso | 8.293 ± 0.366 | 5.977 ± 0.311 | 0.007 ± 0.001 | 3.628 ± 0.061 | 4.346 ± 0.042 | 0.533 ± 0.007 | 3.889 ± 0.061 | 0.781 ± 0.432 |
| Ridge | 8.337 ± 0.312 | 5.97 ± 0.342 | 0.006 ± 0.001 | 3.628 ± 0.060 | 4.343 ± 0.044 | 0.532 ± 0.005 | 3.883 ± 0.061 | 0.681 ± 0.144 |
| Bayesian | 8.318 ± 0.345 | 6.009 ± 0.376 | 0.006 ± 0.001 | 3.628 ± 0.060 | 4.343 ± 0.042 | 0.532 ± 0.005 | 3.884 ± 0.061 | 3.739 ± 0.419 |
| LGBM | 2.834 ± 0.385 | 0.382 ± 0.038 | 0.006 ± 0.001 | **2.266 ± 0.077** | 2.534 ± 0.047 | **0.294 ± 0.006** | 2.582 ± 0.054 | 1.368 ± 0.189 |
| CatBoost | **2.823 ± 0.286** | 0.257 ± 0.028 | 0.001 ± 0.001 | 2.382 ± 0.092 | 2.503 ± 0.029 | 0.295 ± 0.003 | **2.542 ± 0.046** | 1.105 ± 0.153 |
| Tabnet | 4.037 ± 0.561 | 2.643 ± 1.360 | 0.015 ± 0.003 | 3.437 ± 0.123 | 2.723 ± 0.136 | 0.440 ± 0.016 | 2.749 ± 0.022 | 3.316 ± 1.401 |
| NAM | 4.286 ± 0.517 | 2.130 ± 0.254 | 0.041 ± 0.022 | 2.668 ± 0.084 | 3.848 ± 0.033 | 0.374 ± 0.004 | 2.684 ± 0.045 | 3.135 ± 0.199 |
| Ours (shared MLP) | 3.564 ± 0.367 | 0.204 ± 0.03 | 0.001 ± 0.001 | 2.428 ± 0.068 | **2.567 ± 0.072** | 0.393 ± 0.088 | 2.637 ± 0.063 | 0.404 ± 0.148 |
| Ours (mult MLP) | 3.315 ± 0.672 | **0.201 ± 0.021** | **0.001 ± 0.001** | 2.565 ± 0.077 | 2.598 ± 0.123 | 0.422 ± 0.058 | 2.619 ± 0.079 | **0.383 ± 0.179** |

NA represents missing value that indicates the experiments cannot be performed because of the type of model and data size.

For machine learning models, LightGBM model showed the best performance on three datasets, (Concrete strength, Power plant, and California House) while the CatBoost model achieved the best results on two datasets (Protein Structure and FICO) using RMSE evaluation metric. In terms of MEA metric, LightGBM and CatBoost models achieved the performance on two datasets, respectively. Although the neural network-based NAM and TabNet models showed comparable results on most datasets, these models could not achieve superior predictive performances.

In addition, in order to clearly compare baseline models with our proposed model, we measured normalized average RMSE and MAE on all datasets as shown in Figures 2 and 3. Our model is ranked the first and fourth places for the normalized average RMSE evaluation metric (see Figure 2). LightGBM and Catboost models showed the second and

third best predictive performances by achieving 0.428 and 0.426 normalized average RMSE, respectively. Figure 3 displayed the normalized average MAE for all models and we can see that our proposed model consisting of multiple and single MLP models showed the second and third best predictive performance by achieving 0.384 and 0.387 normalized average MAE scores. From experimental results, we can now observe that augmenting linear regression by neural networks showed the state-of-the-art predictive performance without depreciating its interpretability.

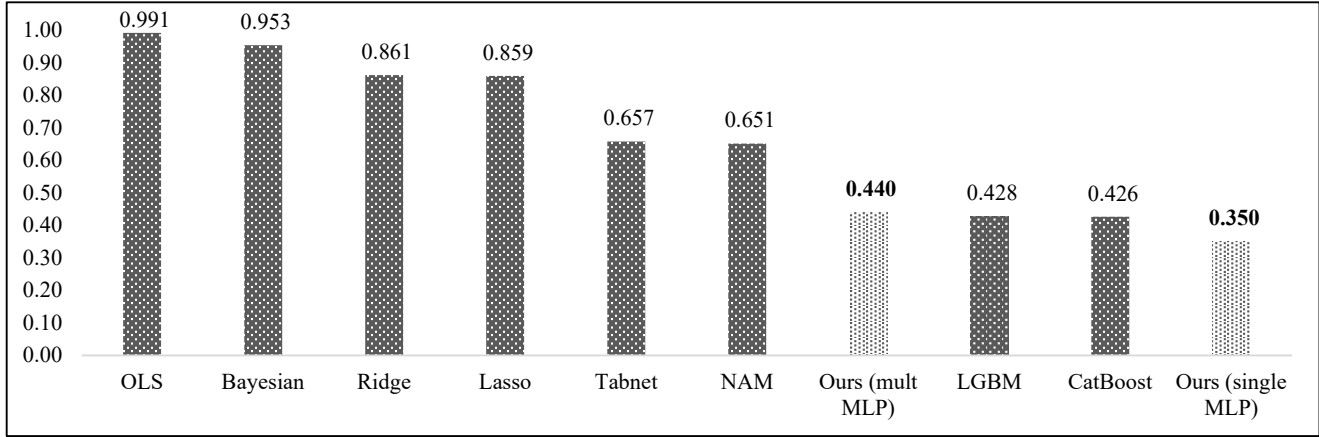

**Figure 2.** Normalized average RMSE for each model.

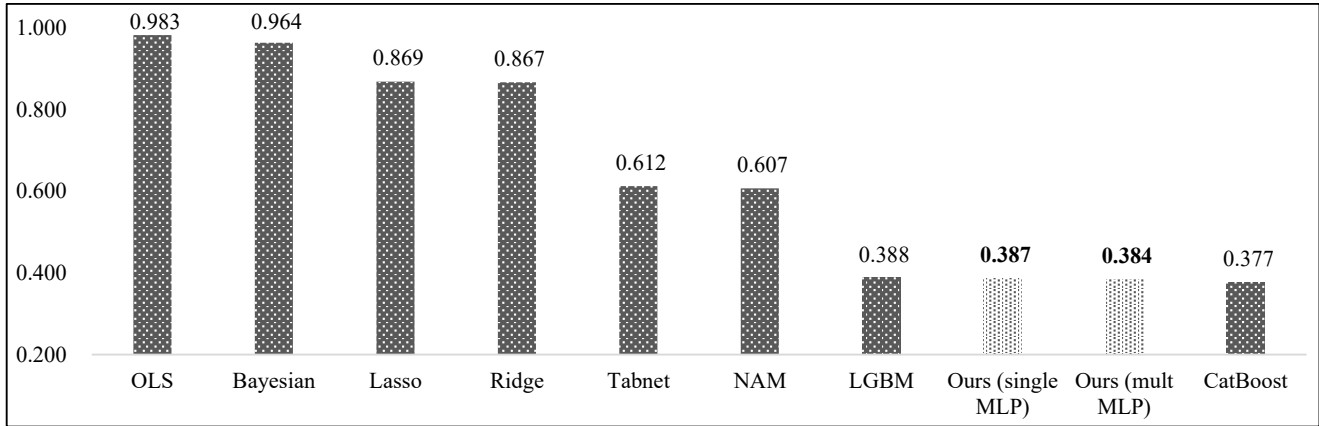

**Figure 3.** Normalized average MAE for each model.

### 4.5. Model Interpretability

We evaluated our proposed model on synthetic and the real-world economic datasets to demonstrate its interpretability. We first aimed to evaluate how our proposed model accurately predicts the known regression coefficients on synthetic datasets. We also trained our model on the real-world economic data to explore the dynamic effect of the central bank policy rate (MN Policy rate) on inflation. We also performed the analysis on $CO_2$ emission data, and our model discovered some interesting explanations between input and target variables, such as a parabolic relationship between $CO_2$ emissions and gross national product (GNP).

### 4.5.1. Model Interpretability on Synthetic Data

In order to evaluate model interpretability, we generated the synthetic datasets based on linear and nonlinear functions. We then trained our proposed model on those datasets to predict the known regression coefficients. The coefficients used to create the synthetic

datasets were derived from a normal distribution rather than using constant coefficients. Based on these known coefficients, we generated the datasets using linear and nonlinear functions; 1. Linear, 2. Quadric, and 3. Summation of multiplication function, which is a summation of multiplication between input variables. The used functions are as the following:

$$y = \sum_{i=0}^{p} \beta_i x_i + \varepsilon \tag{13}$$

$$y = \sum_{i=0}^{p} \alpha_i x_i^2 + \varepsilon \tag{14}$$

$$y = \sum_{i=0}^{p} \alpha_i (x_j + x_k) x_i + \varepsilon \tag{15}$$

where $p$ represents number of variable, $\beta$ and $\alpha$ are parameters used to calculate target variable $y$, $x_i$ is input *i-th* variable, $j$ and $k$ are index for variables (these index should be less than $p$), and $\varepsilon$ is independent, identically distributed random error. For Equation (13), meta-learner in our model should predict $\beta$ as model coefficients. For Equations (14) and (15), the meta-learner in our model will predict $\alpha x_i$ and $\alpha(x_j + x_k)$, which are adaptive regression coefficients.

Based on the above three functions, we generated 10,000 samples of synthetic dataset consisting of six variables.

We then displayed the relationship between known and predicted coefficients for the synthetic data generated by using linear function based on scatter plot in Figure 4. We can see that our proposed model cannot predict the actual coefficients because the known coefficients were randomly derived from normal distribution, and linear regression perfectly predicts target variable.

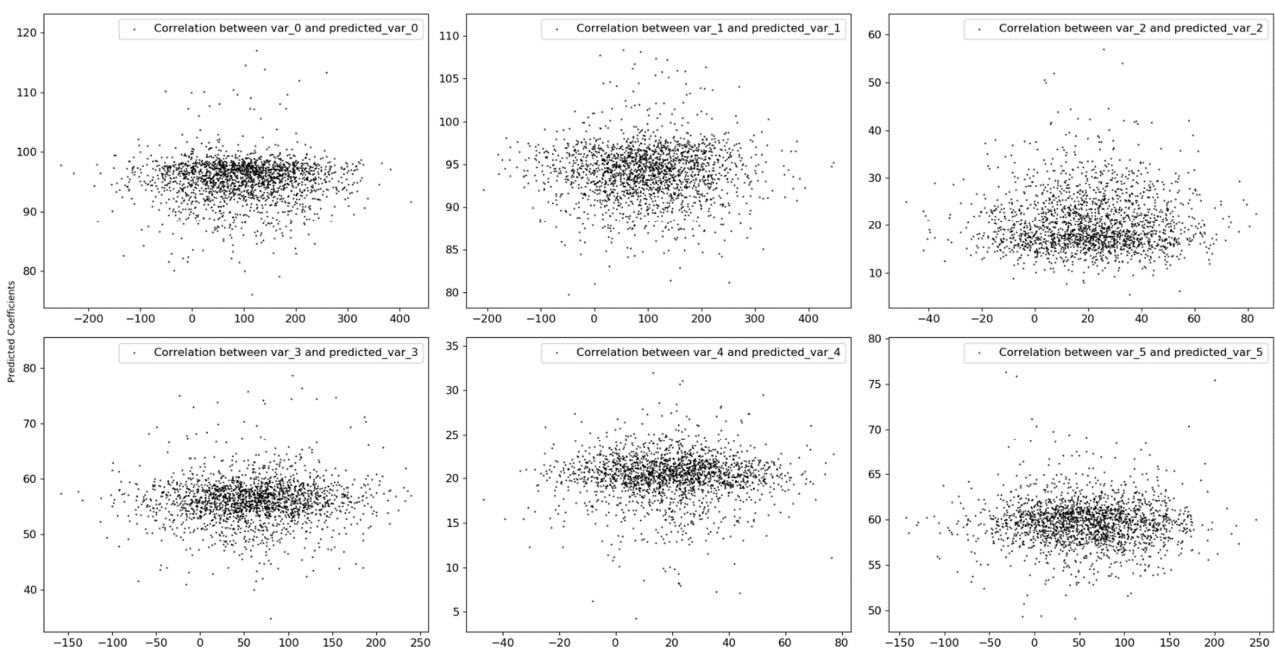

**Figure 4.** The scatter plot between known and predicted coefficients for linear function on test set.

Figures 5 and 6 showed the relationship between known and predicted coefficients for the synthetic data generated by quadric and summation of multiplication functions.

We can see that our model can accurately predict the actual coefficients for the synthetic data generated by using nonlinear functions. Table 4 also presented correlation between known and predicted coefficients for each synthetic data. We now can see that there is no correlation between actual and predicted coefficients for the synthetic dataset generated by linear function. In contrast, for the synthetic datasets generated by quadric and summation of multiplication functions, the actual and predicted coefficients are highly correlated with each other.

**Table 4.** Correlation between known and predicted coefficients for each synthetic data.

| Functions | Coefficients | Known Coefficient-1 | Known Coefficient-2 | Known Coefficient-3 | Known Coefficient-4 | Known Coefficient-5 | Known Coefficient-6 |
|---|---|---|---|---|---|---|---|
| Linear function | Predicted coefficient-1 | **0.010** | 0.006 | 0.002 | −0.015 | −0.022 | 0.024 |
| | Predicted coefficient-2 | 0.023 | **0.008** | 0.033 | −0.013 | −0.007 | 0.006 |
| | Predicted coefficient-3 | 0.003 | 0.011 | **0.012** | 0.017 | 0.010 | −0.002 |
| | Predicted coefficient-4 | 0.000 | −0.032 | 0.047 | **0.041** | −0.005 | 0.004 |
| | Predicted coefficient-5 | −0.032 | 0.020 | −0.036 | −0.025 | **0.004** | −0.031 |
| | Predicted coefficient-6 | −0.005 | 0.035 | −0.010 | −0.012 | −0.007 | **−0.011** |
| Quadric function | Predicted coefficient-1 | **0.999** | −0.005 | −0.014 | −0.054 | 0.001 | 0.049 |
| | Predicted coefficient-2 | −0.001 | **0.999** | 0.028 | 0.005 | 0.009 | 0.032 |
| | Predicted coefficient-3 | −0.020 | 0.023 | **0.999** | −0.043 | 0.033 | 0.015 |
| | Predicted coefficient-4 | −0.049 | 0.018 | −0.030 | **0.997** | 0.034 | −0.045 |
| | Predicted coefficient-5 | −0.007 | 0.007 | 0.028 | 0.026 | **0.999** | −0.034 |
| | Predicted coefficient-6 | 0.055 | 0.040 | 0.019 | −0.034 | −0.027 | **0.999** |
| Summation of multiplication function | Predicted coefficient-1 | **0.890** | 0.891 | −0.035 | −0.075 | 0.890 | 0.443 |
| | Predicted coefficient-2 | 0.869 | **0.869** | 0.099 | 0.010 | 0.869 | 0.579 |
| | Predicted coefficient-3 | −0.290 | −0.290 | **0.575** | −0.117 | −0.291 | −0.216 |
| | Predicted coefficient-4 | −0.793 | −0.793 | 0.328 | **0.566** | −0.794 | −0.493 |
| | Predicted coefficient-5 | 0.961 | 0.961 | −0.099 | −0.134 | **0.961** | 0.664 |
| | Predicted coefficient-6 | 0.892 | 0.892 | 0.031 | 0.034 | 0.892 | **0.894** |

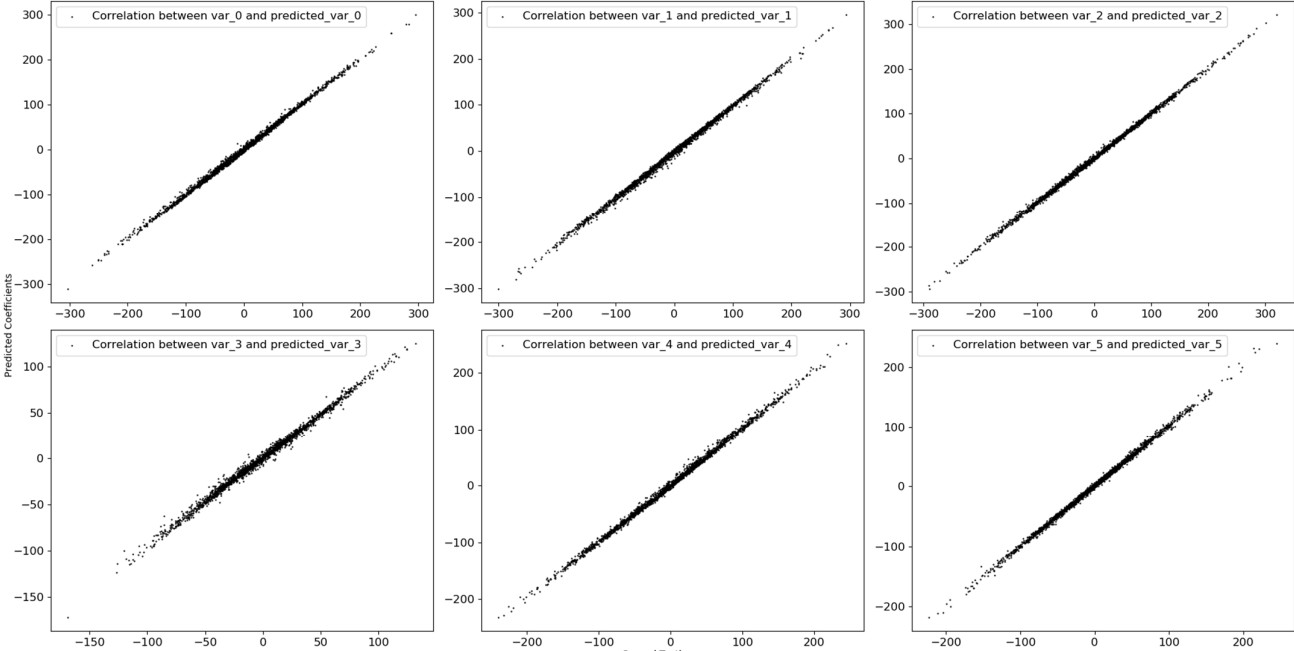

**Figure 5.** The scatter plot between known and predicted coefficients for quadric function on test set.

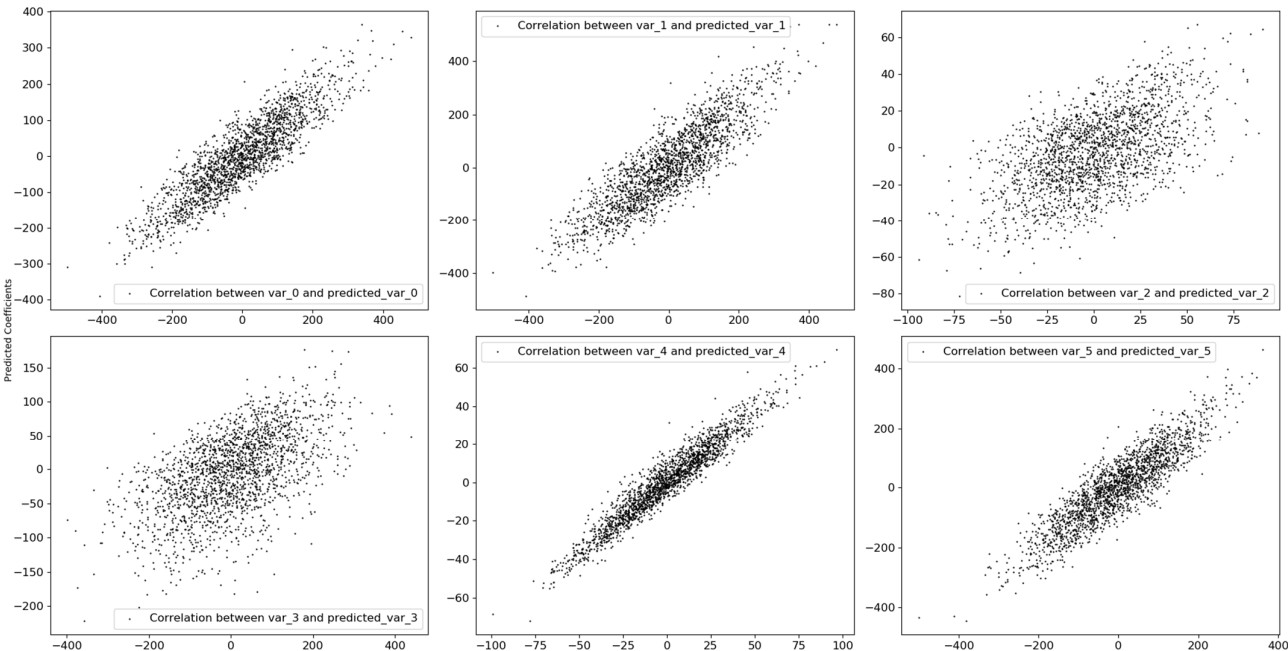

**Figure 6.** The scatter plot between known and predicted coefficients for summation of multiplication function on test set.

Finally, as a result of the experimental analysis on the synthetic datasets, metalearner in our proposed model can explain correlation between input and output variables by approximating the local linear function for each observation.

### 4.5.2. Model Interpretability on Economic Data

We then trained our proposed model on Mongolian economic data to explore the relationship between the central bank policy rate (MN Policy rate) and inflation over time. We retrieved quarterly data from various open sources such as The Central Bank of Mongolia, National Statistics Office of Mongolia, and National Bureau of Statistics of China as shown in Table 5. The data period is from the 4th quarter of 2000 to the 2nd quarter of 2022. For explanatory variables, China's Real Gross Domestic Product (China RGDP, constant price−2015), China's consumer price index (China CPI, price index 2015 = 100), the coal price (Coal price), the oil price (Oil price), the Mongolian Real Gross Domestic Product (MN RGDP, constant price-2015), the budget expenditure of Mongolian (MN Budget Exp), the average wage (MN Average wage), the money supply (MN Money Supply M2), the outstanding loan (MN Loan Outstanding), the central bank policy rate (MN Policy rate) and the terms of trade (MN Terms of trade) are used to forecast Mongolian consumer price index (MN CPI).

**Table 5.** The descriptive statistic and data source for economic data.

| Variables | Period | Yearly Average | Unit | Source |
|---|---|---|---|---|
| China RGDP | 2000Q01–2002Q02 | 51,710.4 | bln YUAN | http://www.stats.gov.cn/english/ accessed on 11 November 2022 |
| China CPI | 2000Q01–2002Q02 | 90.2 | index | http://www.stats.gov.cn/english/ accessed on 11 November 2022 |
| Copper price | 2000Q01–2002Q02 | 5595.8 | USD | https://www.indexmundi.com/commodities/ accessed on 11 November 2022 |
| Coal price | 2000Q01–2002Q02 | 72.9 | USD | https://www.indexmundi.com/commodities/ accessed on 11 November 2022 |

| | | | | |
|---|---|---|---|---|
| Oil price | 2000Q01–2002Q02 | 58.6 | USD | https://www.indexmundi.com/commodities/ accessed on 11 November 2022 |
| MN RGDP | 2000Q01–2002Q02 | 17,304.4 | bln MNT | https://1212.mn/stat.aspx?LIST_ID=976_L05 accessed on 11 November 2022 |
| MN Budget Exp | 2000Q01–2002Q02 | 5195.6 | bln MNT | https://mongolbank.mn/liststatis-tic.aspx?id=0 accessed on 11 November 2022 |
| MN Average wage | 2002Q01–2002Q02 | 568.4 | thou MNT | https://mongolbank.mn/liststatis-tic.aspx?id=0 accessed on 11 November 2022 |
| MN Money Supply M2 | 2000Q01–2002Q02 | 8298.9 | bln MNT | https://mongolbank.mn/liststatis-tic.aspx?id=0 accessed on 11 November 2022 |
| MN Loan Outstanding | 2000Q01–2002Q02 | 7329.0 | bln MNT | https://mongolbank.mn/liststatis-tic.aspx?id=0 accessed on 11 November 2022 |
| MN Policy rate | 2000Q01–2002Q02 | 10.8 | percent | https://mongolbank.mn/liststatis-tic.aspx?id=0 accessed on 11 November 2022 |
| MN Terms of trade | 2000Q01–2002Q02 | 99.4 | index | https://mongolbank.mn/liststatis-tic.aspx?id=0 accessed on 11 November 2022 |
| MN CPI | 2000Q01–2002Q02 | 72.4 | index | https://mongolbank.mn/liststatis-tic.aspx?id=0 accessed on 11 November 2022 |

Before proceeding to OLS estimation, the traditional Augmented Dickey-Fuller (ADF) [48,49] and KPSS [50] unit root tests should be tested to check the hypothesis of stationarity and nonlinearity for all variables [51]. Table 6 shows the result of unit root tests for variables. The optimal lags are selected based on AIC information criterion and maximum lag is equal to 4. The traditional unit root test results showed that the null hypothesis cannot be rejected for most variables; therefore, the variables must be transformed into stationarity. Only the MN Policy rate variable can reject the null hypothesis, so no transformation is needed. Moreover, since this variable is expressed as a percent, we need to transform the other variables to the same level.

**Table 6.** The results of ADF and KPSS unit root test before transformation.

| Variables | Level | | Level | |
|---|---|---|---|---|
| | **ADF Stat** | ***p*-Value** | **KPSS Stat** | ***p*-Value** |
| China RGDP | 1.08 | 1.00 | 1.80 | 0.01 |
| China CPI | 0.04 | 0.96 | 1.59 | 0.01 |
| Coal price | 1.17 | 1.00 | 0.78 | 0.01 |
| Oil price | −1.84 | 0.36 | 0.43 | 0.06 |
| MN RGDP | −0.41 | 0.91 | 1.77 | 0.01 |
| MN Budget Exp | 2.65 | 1.00 | 1.45 | 0.01 |
| MN Average wage | 1.64 | 1.00 | 1.51 | 0.01 |
| MN Money Supply M2 | 3.02 | 1.00 | 1.43 | 0.01 |
| MN Loan Outstanding | 1.70 | 1.00 | 1.53 | 0.01 |
| MN Policy rate | −3.95 | 0.00 | 0.14 | 0.10 |
| MN Term of trade | −1.42 | 0.57 | 1.01 | 0.01 |
| MN CPI | 2.89 | 1.00 | 1.59 | 0.01 |

After transforming the variables, as shown in Table 7, ADF and KPSS test results showed that the null hypothesis cannot be rejected for d4log(China RGDP) and d1log(MN Money Supply M2) variables. On the other hand, the variable d4log(China CPI) was highly correlated with the variable d4log(MN RGDP) and the variable d1log(MN Money Supply M2) was also highly correlated with the variable d1log(MN Loan Outstanding), so we excluded these variables from the regression model.

**Table 7.** The results of ADF and KPSS unit root test after transformation.

| Variables | Transformed | | Transformed | | Formula |
|---|---|---|---|---|---|
| | **ADF Stat** | ***p*-Value** | **KPSS Stat** | ***p*-Value** | |
| d4log(China RGDP) | −2.26 | **0.18** | 0.79 | **0.01** | $\log(x_t) - \log(x_{t-4})$ |
| d1log(China CPI) | −4.32 | 0.00 | 0.09 | 0.10 | $\log(x_t) - \log(x_{t-1})$ |
| d1log(Coal price) | −7.37 | 0.00 | 0.13 | 0.10 | $\log(x_t) - \log(x_{t-1})$ |
| d1log(Oil price) | −7.64 | 0.00 | 0.09 | 0.10 | $\log(x_t) - \log(x_{t-1})$ |
| d4log(MN RGDP) | −2.65 | 0.08 | 0.35 | 0.10 | $\log(x_t) - \log(x_{t-4})$ |
| d4log(MN Budget Exp) | −2.82 | 0.06 | 0.20 | 0.10 | $\log(x_t) - \log(x_{t-4})$ |
| d1log(MN Average wage) | −3.37 | 0.01 | 0.31 | 0.10 | $\log(x_t) - \log(x_{t-1})$ |
| d1log(MN Money Supply M2) | −3.29 | 0.02 | 0.54 | **0.03** | $\log(x_t) - \log(x_{t-1})$ |
| d1log(MN Loan Outstanding) | −2.86 | 0.05 | 0.93 | **0.01** | $\log(x_t) - \log(x_{t-1})$ |
| MN Policy rate | −3.70 | 0.00 | 0.15 | 0.10 | $x_t$ |
| d1log(MN Term of trade) | −4.18 | 0.00 | 0.06 | 0.10 | $\log(x_t) - \log(x_{t-1})$ |
| d1log(MN CPI) | −3.28 | 0.02 | 0.10 | 0.10 | $\log(x_t) - \log(x_{t-1})$ |

For the variables for which traditional unit root tests were accepted, we considered that these variables could affect inflation and are included in the OLS estimation.

A total of 78 quarters from the 1st quarter of 2001 to the 2nd quarter of 2020 are used as the training set, and 8 quarters from the 3rd quarter of 2020 to the 2nd quarter of 2022 are considered as a test set.

From the result of the OLS regression (Tables 8 and 9), we can see a negative relationship between the central bank policy rate (MN Policy rate) and inflation d1log(MN CPI). Expressly, in the Mongolian economy, inflation tends to fall by 0.0025 percent if the central bank raises the policy rate by one percent.

**Table 8.** Estimated coefficients of OLS regression on economic data.

| Variables | Coefficients | *p*-Value |
|---|---|---|
| d4log(China RGDP) | | |
| d1log(China CPI) | 1.183 *** | 0.000 |
| d1log(Coal price) | 0.002 | 0.936 |
| d1log(Oil price) | 0.006 | 0.681 |
| d4log(MN RGDP) | −0.0529 * | 0.062 |
| d4log(MN Budget Exp) | 0.0419 * | 0.071 |
| d1log(MN Average wage) | 0.1117 ** | 0.039 |
| d1log(MN Money Supply M2) | | |
| d1log(MN Loan Outstanding) | 0.1324 *** | 0.003 |
| MN Policy rate | −0.0025 ** | 0.029 |
| d1log(MN Term of trade) | 0.000 | 0.803 |
| d1log(MN CPI) | 0.106 | 0.331 |
| Intercept | −0.033 ** | 0.044 |
| R-squared | 0.403 | |
| Adj. R-squared | 0.314 | |

\*\*\*, \*\*, \* significantly different from zero at 0.01, 0.05, and 0.1 level, respectively.

**Table 9.** Regression diagnostics result on economic data.

| Statistic Test | Value |
|---|---|
| Jarque–Bera test | |
| Jarque–Bera | 0.543 |
| Chi² two-tail prob | 0.762 |
| Skew | 0.204 |
| Kurtosis | 2.980 |
| Omnibus normtest | |
| Chi² | 0.721 |
| Two-tail probability | 0.697 |
| Breusch–Pagan Lagrange Multiplier test | |
| Lagrange multiplier | 0.546 |
| *p*-value | 0.128 |
| f-value | 0.857 |
| f *p*-value | 0.576 |
| Durbin–Watson test | |
| Durbin–Watson test | 1.974 |

Table 10 also showed the predictive performance results, and we can see that our proposed model reduces the error of the OLS model by an average of 0.0107 units of inflation, or 30.7 percent.

**Table 10**. The predictive performance on economic test set.

| Model | RMSE | MAE |
|---|---|---|
| Ours | 0.0271 | 0.0242 |
| OLS | 0.03943 | 0.0349 |
| Reduction | −0.0123 | −0.0107 |
| Reduction (%) | −31.3% | −30.7% |

Another advantage of our proposed model is to capture how the regression coefficients change over time. In macroeconomics, the impact of explanatory variables on inflation can be changed over time depending on the economic situation [52]. Therefore, our model can be suitable for developing economic models rather than linear regression.

Figure 7 showed how the central bank's policy rate affects inflation over time. As we know, there was a global economic crisis in the period between 2008 and 2010. During this period, we can see that the effect of the central bank policy rate on inflation is high. In other words, if the central bank raises the policy rate by one percent, inflation tends to fall by more than 0.03 percent. On the contrary, between 2010 and 2012, Mongolia's largest copper mine called Oyu Tolgoi started and the Mongolian economy was extremely grown. At the same time, the effect of the central bank's policy rate on inflation weakened.

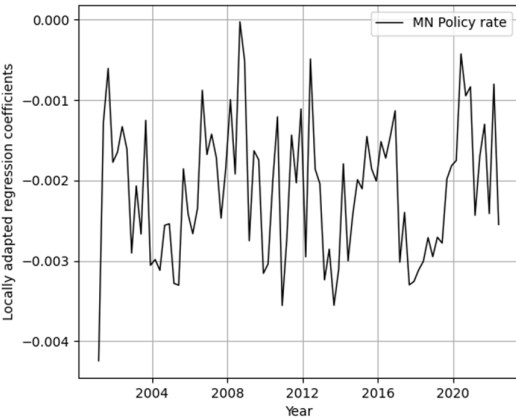

**Figure 7.** The impact of the central bank's policy rate on inflation over time.

The estimated regression coefficient from the linear regression is consistent with the economic theory for MN Policy rate. We can now see that our proposed model showed how the effect of MN Policy rate changes dynamically on inflation while keeping this consistency.

### 4.5.3. Model Interpretability on $CO_2$ Emission Data

In this section, we considered real-world $CO_2$ emission dataset and examined the link between $CO_2$ emission and gross national product (GNP) [49,50]. We explored the relationship between the $CO_2$ and GNP based our proposed model.

The source of the dataset is the official web page of Our World in Data (https://ourworldindata.org/, accessed on 1 November 2022) and the data between 1990 and 2015 are the training and data in 2016 are the test set (see Table 11). In addition, we also performed ADF and KPSS tests on $CO_2$ emission dataset as shown in Table 12, and ADF and KPSS test results showed that the null hypothesis can be accepted for logarithm scale of both variables. Furthermore, to investigate the link between $CO_2$ emission and GNP, we estimated two different regression equations as follows:

$$log(CO_2) = \beta_0 + \beta_1 \cdot log(GNP) \tag{16}$$

$$log(CO_2) = \beta_0 + \beta_1 \cdot log(GNP) + \beta_2 \cdot log(GNP)^2 \tag{17}$$

**Table 11.** The descriptive statistic for CO2 emission data.

| Variables | #Observation | Mean | STD | Min | Max |
|---|---|---|---|---|---|
| $CO_2$ | 4441 | 0.559 | 1.714 | (4.535) | 4.248 |
| GNP | 4441 | 8.942 | 1.243 | 5.063 | 11.960 |
| $GNP^2$ | 4441 | 81.508 | 22.051 | 25.630 | 143.030 |

\# The abstract of number of observations.

**Table 12.** The results of ADF and KPSS unit root test for $CO_2$ emission data.

| Variables | Level | | Level | |
|---|---|---|---|---|
| | ADF Stat | *p*-Value | KPSS Stat | *p*-Value |
| GNP | −0.86 | 0.80 | 0.63 | 0.02 |
| $CO_2$ | −1.47 | 0.55 | 0.50 | 0.04 |
| log(GNP) | −4.38 | 0.00 | 0.08 | 0.10 |
| $log(GNP)^2$ | −4.38 | 0.00 | 0.08 | 0.10 |
| $log(CO_2)$ | −4.49 | 0.00 | 0.25 | 0.10 |

In general, assuming that there are positive linear and negative parabolic relationships between $CO_2$ emission and GNP. Theoretically, the Environmental Kuznets Curve (EKC) hypothesis postulates an inverted-U-shaped relationship between $CO_2$ emission and GNP [53,54].

Our estimates of Equations (16) and (17) are reported in Table 13 and regression diagnostics result is presented in Table 14. The regression coefficients are consistent with EKC hypothesis. We then trained our model on these two OLS results and reported the prediction performance in Table 15. Our model showed slightly better performance than both OLS results. Finally, we captured the relationship between $CO_2$ emission and GNP.

**Table 13.** Estimated coefficients of OLS regression on $CO_2$ emission data.

| Variables | Linear Term | Linear and Quadratic Terms |
|---|---|---|
| Intercept | −10.76 *** | −18.81 *** |
| log(GNP) | 1.27 *** | 3.13 *** |
| log(GNP)$^2$ | | −0.105 *** |
| R-squared | 0.839 | 0.85 |
| Prob (F-statistic) | 0.00 | 0.00 |

*** significantly different from zero at 0.01 level.

**Table 14.** Regression diagnostics result on $CO_2$ emission data.

| Statistic Test | Value |
|---|---|
| Jarque–Bera test | |
| Jarque–Bera | 2.577 |
| Chi$^2$ two-tail prob | 0.276 |
| Skew | 0.128 |
| Kurtosis | 3.562 |
| Omnibus normtest | |
| Chi$^2$ | 2.748 |
| Two-tail probability | 0.253 |
| Breusch–Pagan Lagrange Multiplier test | |
| Lagrange multiplier | 2.194 |
| *p*-value | 0.334 |
| f-value | 1.091 |
| f *p*-value | 0.338 |
| Durbin–Watson test | |
| Durbin–Watson test | 2.097 |

**Table 15.** The prediction performance on $CO_2$ emission test dataset.

| Model | RMSE | MAE | R-Squared |
|---|---|---|---|
| LAEN with linear term | 0.591 | 0.482 | 0.851 |
| OLS with linear term | 0.607 | 0.483 | 0.843 |
| LAEN with linear and quadratic terms | 0.593 | 0.481 | 0.85 |
| OLS with linear and quadratic terms | 0.598 | 0.488 | 0.848 |

Figure 8 showed the influence of GNP (left) and intercept (right) on $CO_2$ emission for Equation (16). We can easily see that GNP and intercept are parabolic with $CO_2$ emission. When GNP goes up to 9.16, it intensively increases $CO_2$ emission, then when GNP is higher than 9.16 its effect on $CO_2$ emission starts to decrease. For intercept, average $CO_2$ emission increases up to a certain level as GNP goes up; after that, it decreases. In Equation

(17), we added the quadratic term of GNP as an explanatory variable, and the predictive performance of OLS is improved. Although the predictive performance of our model has not changed much, its interpretability is shifted as shown in Figure 9. We can now see that the parabolic relationship between $CO_2$ and GNP on Equation (17) has transformed to linear. Our model can also measure how much $CO_2$ will change due to the change in GNP for each country.

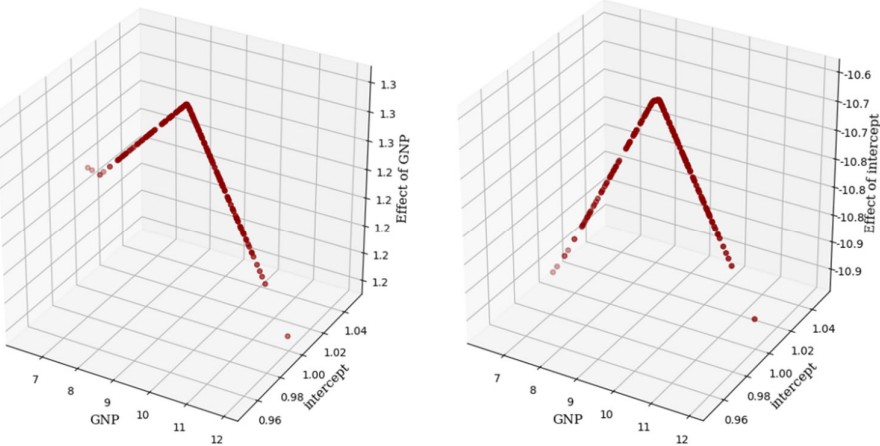

**Figure 8.** The relationship between GNP, intercept and the estimated coefficients obtained by our proposed model.

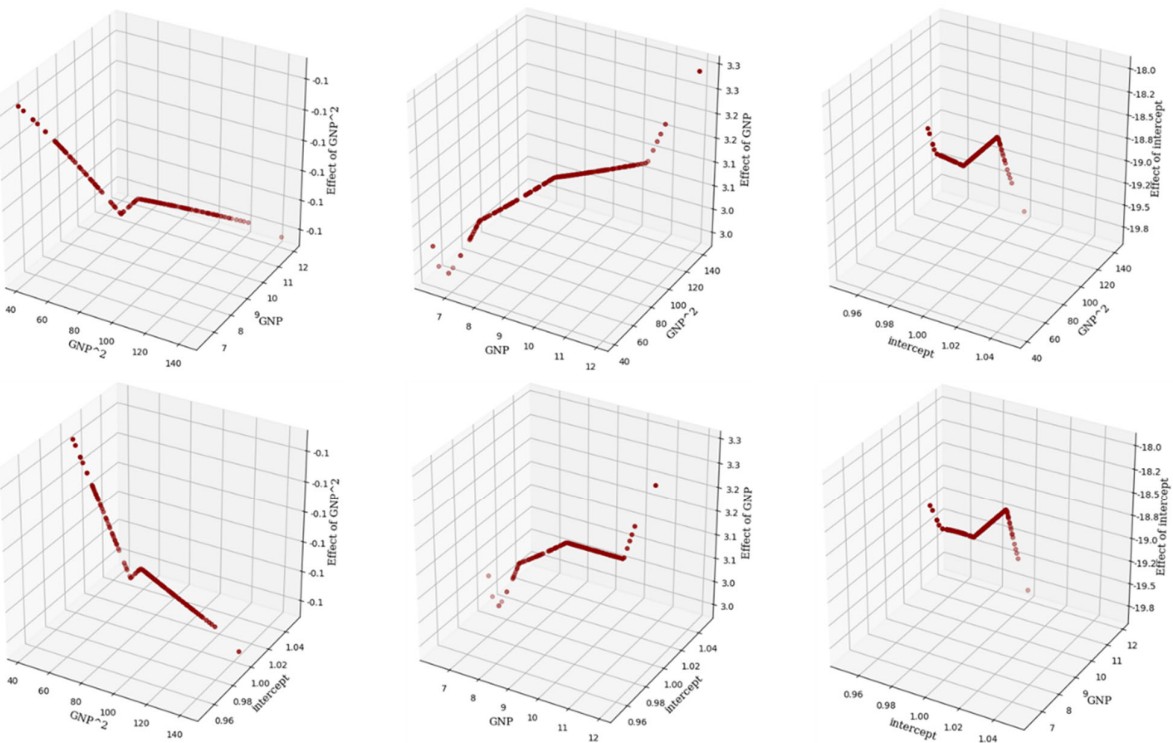

**Figure 9.** The relationship between GNP, the quadratic term of GNP, intercept and the estimated coefficients obtained by our model.

## 5. Conclusions

In this work, we aimed to create a white-box model for regression tasks on tabular data. In order to provide both high predictive accuracy and explainability, we proposed a novel locally adaptive interpretable regression model augmented by neural networks. The proposed model relies on two key aspects. First, a base-learner should be a simple interpretable model. In this work, we obtain our base-learner using OLS regression and its statistical properties. Second, we use neural networks as our meta-learner to re-parameterize our base-learner to produce a local interpretable linear model for each observation. We can locally explain the relationship between input and output variables based on the adapted local regression coefficients. We evaluate the predictive performance and interpretability of our proposed model on several tabular datasets. Experimental results showed that our model greatly improved the predictive performance of OLS regression after being augmented by neural networks. Our model is ranked first by the normalized average RMSE and second by the normalized average MAE from experimental results.

In addition, in order to evaluate model explainability, we perform additional experiments on the synthetic, economic, and $CO_2$ emission datasets. For the synthetic data generated by non-linear functions, our proposed model can explain the relationship between input and output features by approximating a local linear function for each observation. We then perform an analysis of economic time-series data, and our model explores the dynamic relationship between input and output variables. As a result, we have observed that the impact of central bank policy rates on inflation tends to weaken during a recession and rises during an expansion, consistent with the economic theory. Lastly, we applied our model to $CO_2$ emission data, and our model discovers some interesting explanations between input and target variables, such as a parabolic relationship between $CO_2$ emissions and gross national product (GNP).

We believe that our proposed model can be applicable for many real-world domains where data type is tabular and interpretable models are required.

**Author Contributions:** Conceptualization, L.M., P.V.H. and K.H.R.; methodology, L.M. and T.M.; software, L.M.; validation, P.V.H., N.T.-U., and J.-E.H.; formal analysis, L.M., and K.H.R.; investigation, L.M.; resources, K.H.R.; data curation, L.M.; writing—original draft preparation, L.M., and K.H.R.; writing—review and editing, P.V.H., N.T.-U., and J.-E.H.; visualization, L.M.; supervision, J.-E.H., and K.H.R.; project administration, K.H.R.; funding acquisition, K.H.R. All authors have read and agreed to the published version of the manuscript.

**Funding:** This work was supported by the Basic Science Research Program through the National Research Foundation of Korea (NRF) funded by the Ministry of Science, ICT, and Future Planning under Grant No. 2020R1A2B5B02001717 in Korea.

**Institutional Review Board Statement:** Not applicable.

**Informed Consent Statement:** Not applicable.

**Data Availability Statement:** Data available in a publicly accessible repository.

**Acknowledgments:** This work has been done partially while Lkhagvadorj Munkhdalai visited in the Biomedical Engineering Institute, Chiang Mai University.

**Conflicts of Interest:** The authors declare no conflict of interest.

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
