# Peer review of "Neural Network-Augmented Locally Adaptive Linear Regression Model for Tabular Data"

_sustainability, doi:10.3390/su142215273_

Round 1

Reviewer 1 Report

Dear Editor and Dear Authors,

I reviewed the paper and following issues should be addressed. 

1. Combining regression analysis with neural networks is neither new nor novel. There is an existing literature on it for the last 3-4 decays. 

2. Eq. 3 is not readable. 

3. The proposed model given in figure 1 depicts the proposed model and it seem preliminary. The contribution is not clear. It should be better presented. 

4. Check the formula of MLP at eq: 4. I suggest Bishop 1995 as a reference.

5. Eq. 5 is not clearly visible. This is the second eq. in which the parameters cannot be clearly seen. Same for Eq. 6.(also for the rest).

6. Equations and model has no error terms. If there would be no error terms, how would standard errors be produced which you show in your equations? Models should include error terms. 

7. Loss function cannot be read and cannot be understood. 

8. Why do the epsilons not have t, time index while the variables have?

9. On part 4.5.2: Data sources are not specifically stated variable by variable. It should be. And, some varaibles are in logarithms while others are not. Why? I suggest logarithmic transformation for all. 

10. Even though the paper has an intention to combine regression analysis a la econometrics with MLP, the econometric methodology is not followed. In addition to previous critique, no descriptive statistics table is reported, no distributional properties are reported, no unit root tests are conducted. How do we know that the regression result is not spurious regression? In addition, same for after estimation: after mmodel estimations, no diagnostics tests are reported, no autocorrelation testing, no heteroskedasticity testing, no paremeter stability testing. For a paper that has distributional concerns, such factors should be carefully conducted. Note: this would necessitate revision of the estimations and therefore the whole application section. 

11. On table 6: Comparing with a NN augmented model with simple OLS is not enough. Further, why not compared with other models proposed by other researchers for the last decays with similar intentions, intentions to combine neural networks with regressions. These attempts to stochastic modelling of NN models with regressions should be included to the literature, intro. and to the table 6 (one or some models can be incorporated to be compared with your model to this table). Relevant changes should be made in the method section (stating these models and giving references to them by discussing your models' differences and similarities.  

12. The following references should be added to references. In addition, they should be used in conjunction with the critiques stated above and the corrections to be done.  

* Improving forecasts of GARCH family models with the artificial neural networks: An application to the daily returns in Istanbul Stock Exchange

* Modeling Markov switching ARMA-GARCH neural networks models and an application to forecasting stock returns

* TAR-cointegration neural network model: An empirical analysis of exchange rates and stock returns   

* Modeling Markov Switching ARMAGARCH Neural Networks Models and an Application to Forecasting Stock Returns

* Learning in Artificial Neural Networks: A Statistical Perspective

* Markov-switching vector autoregressive neural networks and sensitivity analysis of environment, economic growth and petrol prices

* Building neural network models for time series: a statistical approach

13. The paper is submitted to Sustainability however the paper does not have the use of the term sustainability. Sustainability should be integrated to the model since it is a major concern of this special issue the paper is submitted to. I suggest that authors emphasize sustainability and the impact of their model on it. The used datasets are also a part of a repository. However, if the variables used in this paper are checked, which ones are related to sustainability? Some but certainly not the majority (Ex. Energy can be though to be so). I suggest addition of variables such as CO2 and carbon footprint. 

14. Regarding the critique above: The authors of course can think of submitting the paper to another neural networks or machine learning focused journal since it would be more appropriate. This decision is not up to me. But my suggestions are above and as seen, my suggestion is addressing the critiques and making the corrections also by emphasizing sustainability issue. The paper has potential and has merits. I believe that the paper will be more suitable to sustainability after the corrections. Some corrections necessitate major revisions since it necessitates revision of the whole empirics in addition to other sections. The literature should also be revised and so is conclusion after the corrections being made. Lastly, the last sentence of conclusion is a very general suggestion, instead of this, the paper should have more direct and focused suggestions but this is not presented in this conclusion.  

Author Response

Dear Reviewer,

Thank you for reviewing our manuscript and giving us valuable comments. According to your comments, we have improved our manuscript. Please check the new version of our manuscript and responses. 

Reviewer 2 Report

The paper describes an interpretable neural network-based regression model for tabular data. While this is an interesting subject belonging to a hot research area, some clarifications show be added to the current form of the paper to improve its added value:

- the classical regression model with OLS estimates has certain statistical properties that are very well studied; what about the proposed model? is the estimated y biased or not? Is the estimator a the minimum variance unbiased estimator or not? Is the estimator consistent or not? For a model to be interpretable, as authors want, these properties should be know. Otherwise the proposed model it is like any other ML model.

-why the updated coefficients are in the form of beta_i + t * se(beta_i)? Why not beta_i - t * se(beta_i) or any other form?

-for the neural network to work, we need to know the "true" outputs, to be able to compute the error and update the weights. The questions is: which are the "true" outputs used by the network compare to compare the actual outputs with? NN is  supervised method. From the actual form of paper it is not clear which are the true values, a reader can only guess.

- the properties of the data sets should be presented somehow, if not in the main paper, maybe in an annex.

Author Response

(The authors gave the same response as above.)

Round 2

Reviewer 1 Report

1. Eq 3, the confidence intervals, s.e.(beta), beta should have a hat.
2. In many equations, there are some Korean letters on top of Greek anotations that are on top of each other. Is it due to authors' writing of equations or there has been a problem in PDF conversion? It should be corrected in many equations including Eq 3 and 6 as examples.
Note that these points were made but still at this round, they exist. Authors also stated that they corrected them in the rebutal. They should recheck the equations for such problems. In fact, the rebutal letter of authors are full of similar problems in terms of Korean letters on top of some betas...
3. Equations 13,14,15 are too large in terms of font.
4. Chine GDP and China CPI based inflation calculations. Is GDP real or nominal? If real, constant prices at which year? CPI is which CPI, I mean, what is the basis year? These details should be present.
5. In equations, all se(beta)'s should have hats on top of betas.
6. Point 10 is not met. The explanation is not adequate. If the references sent in the previos round are checked, unit root testing is a necessity in the method both econometric and also for neural nets. Therefore, authors should take it seriously and make the necessary corrections. Further, why are some variables in logarithms while others are not? This was asked before. They all should be in the same basis, in logarithms. If the reason is using growth rates and inflation rates at the same time in addition to stock variables, still this does not make sense. Taking the logarithms of CPI as LCPI = Log(CPI) and then checking for unit roots might lead to (most probably) that the series is not stationary. Then, first differencing will occur if series is integrated of order 1. After taking the first difference of the LCPI, the obtained DLCPI, which is DLCPI=D(LCPI) is equivalent to the inflation rates anyways. Therefore, it is still my opinion that the analysis should be revised with this respect. This will necessitate recalculation of the rest of tables such as forecast comparisons.
7. All critiques above also hold for the newly added section that deals with CO2 now. Unit root testing is a very serious concept that cannot be overlooked.

I am sorry but revisions are necessary again however, though authors addressed many critiques, major ones are still not addressed in this version of the paper which is still not satisfactory.  

Author Response

(The authors gave the same response as above.)

Reviewer 2 Report

The authors have made enough changes according to the initial review.

Author Response

Dear Reviewer,

Thank you for reviewing our manuscript and giving us valuable comments. 

Best Regards,

Authors

Round 3

Reviewer 1 Report

Authors have gone over the critiques I directed at the second round. I observe that they made the corrections carefully. I am happy with this revised version of this paper. Congrats to the authors.